# Incorporation of tetracarboxylate ions into octacalcium phosphate for the development of next-generation biofriendly materials

Taishi Yokoi [1,2 ✉], Tomoyo Goto [3], Mitsuo Hara [4], Tohru Sekino [3], Takahiro Seki[4], Masanobu Kamitakahara[5], Chikara Ohtsuki[6], Satoshi Kitaoka[2], Seiji Takahashi[2] & Masakazu Kawashita [1]

Octacalcium phosphate (OCP; $Ca_8(HPO_4)_2(PO_4)_4 \cdot 5H_2O$) is a precursor of hydroxyapatite found in human bones and teeth, and is among the inorganic substances critical for hard tissue formation and regeneration in the human body. OCP has a layered structure and can incorporate carboxylate ions into its interlayers. However, studies involving the incorporation of tetracarboxylic and multivalent (pentavalent and above) carboxylic acids into OCP have not yet been reported. In this study, we investigate the incorporation of pyromellitic acid (1,2,4,5-benzenetetracarboxylic acid), a type of tetracarboxylic acid, into OCP. We established that pyromellitate ions could be incorporated into OCP by a wet chemical method using an acetate buffer solution containing pyromellitic acid. The derived OCP showed a brilliant blue emission under UV light owing to the incorporated pyromellitate ions. Incorporation of a carboxylic acid into OCP imparted new functions, which could enable the development of novel functional materials for biomedical applications.

[1] Institute of Biomaterials and Bioengineering, Tokyo Medical and Dental University (TMDU), 2-3-10 Kanda-Surugadai, Chiyoda-ku, Tokyo 101-0062, Japan. [2] Materials Research and Development Laboratory, Japan Fine Ceramics Center, 2-4-1 Mutsuno, Atsuta-ku, Nagoya 456-8587, Japan. [3] The Institute of Scientific and Industrial Research, Osaka University, 8-1 Mihogaoka, Ibaraki 567-0047, Japan. [4] Department of Molecular and Macromolecular Chemistry, Graduate School of Engineering, Nagoya University, Furo-cho, Chikusa-ku, Nagoya 464-8603, Japan. [5] Graduate School of Environmental Studies, Tohoku University, 6-6-20 Aoba, Aramaki, Aoba-ku, Sendai 980-8579, Japan. [6] Department of Materials Chemistry, Graduate School of Engineering, Nagoya University, Furo-cho, Chikusa-ku, Nagoya 464-8603, Japan. ✉email: yokoi.taishi.bcr@tmd.ac.jp

Octacalcium phosphate (OCP; $Ca_8(HPO_4)_2(PO_4)_4 \cdot 5H_2O$) is an inorganic substance that is critical for hard tissue formation and regeneration in the human body. Hydroxyapatite (HAp; $Ca_{10}(PO_4)_6(OH)_2$) is a well-known inorganic mineral present in the bones and teeth of mammals, including humans[1–4]. OCP is regarded as a precursor of HAp, and is found in hard tissues[5,6]. Additionally, because of its excellent affinity towards bone tissues, OCP is used in surgery as a filling material for bone repair in the case of defects caused by cancer or vehicular accidents[7–9]. OCP is also present in human dental and urinary calculi[10]. Therefore, a deeper understanding of the physicochemical and biological properties of OCP is essential for the development of novel medical devices. In addition, understanding the pathological calcification of OCP in the human body enables the development of methods to suppress this process, resulting in an improved quality of life.

OCP exhibits a unique crystallographic property. It has a layered structure composed of apatitic and hydrated layers[11,12]. In 1983, Monma[13] fortuitously discovered that hydrogen phosphate ions ($HPO_4{}^{2-}$) in the hydrated layers of OCP could be substituted by an organic acid, namely succinic acid ($HOOC(CH_2)_2COOH$), which is a type of saturated dicarboxylic acid. Further studies demonstrated the incorporation of a variety of dicarboxylate ions into OCP[14–21]. It has been reported that precisely controlling the layered structure of OCP at the molecular level would enable the application of dicarboxylate-ion-incorporated OCP as a novel functional material for bone repair[22] and as a specific adsorbent for aldehydes[23]. In addition, previous studies suggested the presence of a similar layered structure of OCP that incorporated a type of tricarboxylate ion (citrate ion) in bone minerals[24]. Therefore, OCP containing a variety of incorporated carboxylate ions has attracted considerable interest from researchers in various fields.

Although the process of carboxylate ion-incorporation in OCPs is similar to the intercalation of inorganic layered compounds, the two processes differ greatly in terms of guest selectivity. Generally, for inorganic layered compounds, the possibility of intercalation is governed by the charge of the guest molecules[25–29]. Although OCP can incorporate carboxylate ions, there have been no reports on the incorporation of organic sulphate and phosphate ions. Only dicarboxylate ions, with a few exceptions such as the citrate ion, which is a type of tricarboxylate ion, can be incorporated into OCP[30,31]. There are no reports on the incorporation of tetracarboxylic acids and carboxylic acids with multiple valences (pentavalent and above). The incorporation of tetravalent or higher-valence carboxylate ions into OCP was not realised for more than 37 years since the findings on the first incorporation of carboxylate ions into OCP were reported[13]. This indicates that the incorporation of tetracarboxylate ions into OCP is synthetically prohibitively difficult; however, we have discovered an effective approach for achieving this incorporation.

We found that a type of dicarboxylic acid exists in the OCP interlayer as a dimer, which could behave as a tetracarboxylic acid. As the incorporated carboxylate ions are parallel to the *a*-axis of OCP, the (100) interplanar spacing ($d_{100}$) of OCP strongly depends on the molecular size of the incorporated carboxylate ions[32]. Base on this, phthalic and isophthalic acid were incorporated into OCP[33]. The $d_{100}$ value of OCP with the incorporated isophthalate ions was expected to be larger than that of OCP with phthalate ions, based on the molecular structure of the carboxylate ions. However, contrary to this expectation, the $d_{100}$ value of OCP with the incorporated phthalate ions (2.37 nm) was larger than that of OCP with the isophthalate ions (2.30 nm)[33]. This implied that the phthalate ions in the hydrated layer of OCP were present as a dimer owing to π-π interactions. Figure 1a and b shows schematic illustrations of the expected structures of the incorporated isophthalate and phthalate ions, respectively. As the phthalic acid dimer has a similar structure to that of pyromellitic acid (1,2,4,5-benzenetetracarboxylic acid, the molecular structure of which is shown as Fig. 1c), the incorporation of pyromellitate ions into OCP could be realised, despite pyromellitic acid being a tetracarboxylic acid.

In this study, to demonstrate that tetracarboxylate ions can be incorporated into OCP, we synthesised OCP with incorporated pyromellitate ions, which was confirmed by crystal-chemical, compositional, and chemical structural analyses. In addition, as recent studies reported the fluorescence of OCP with incorporated aromatic carboxylate ions[34], we studied the fluorescent properties of OCP with incorporated pyromellitate ions. If we can impart fluorescent properties on OCP, it would enable the development of a novel theranostic material enabling bone repair and fluorescence diagnosis.

## Results

**Optimising synthesis conditions: Effects of pyromellitic acid concentration on the reaction solution.** OCP with incorporated pyromellitate ions was synthesised by the hydrolysis of dicalcium phosphate dihydrate (DCPD; $CaHPO_4 \cdot 2H_2O$) in an acetate buffer containing pyromellitic acid. The advantages of this synthetic method are the pH stability and relatively low calcium (Ca) ion concentration during the synthesis process. Both the crystalline phases of calcium phosphates synthesised by wet processes, and the dissociation states of pyromellitate ions strongly depend on the pH of the reaction solution. Therefore, pH stability during synthesis is favourable for the precise synthesis of OCP with incorporated pyromellitate ions. A relatively low Ca ion concentration is achieved owing to the lower solubility of DCPD compared to α-tricalcium phosphate, which is conventionally used as a starting material for synthesising OCP with incorporated carboxylate ions. This is favourable for avoiding the formation of poorly soluble calcium carboxylate salts, which are represented by calcium pyromellitate in this case.

The synthesis conditions of OCP with incorporated pyromellitate ions were optimised by varying the pyromellitic acid concentrations and pH of the reaction solution. Figure 2 shows the powder X-ray diffraction (XRD) patterns of the samples synthesised in reaction solutions with various pyromellitic acid concentrations at an initial solution pH of 5.5. The samples were named according to the following convention, wherein, e.g., Pyro-10 indicates the sample synthesised in a reaction solution containing 10 mol m$^{-3}$ of pyromellitic acid. The reflection peaks of the OCP phase were observed for all samples. The crystalline phase of Pyro-0 was plain OCP and did not contain carboxylate ions. The powder diffraction file (PDF) #01-074-1301 was used to identify plain OCP. For Pyro-1, the characteristic 100 reflection peak of OCP was detected at $2\theta = 4.7°$, along with an additional overlapping peak forming a tail at the low-angle side of the 100 reflection peak. When using higher pyromellitic acid concentrations (5 to 100 mol m$^{-3}$), the 100 reflection peak of OCP was detected at 3.8°, which is a lower angle than that of plain OCP. This peak shift was caused by the incorporation of pyromellitate ions into the OCP crystal. In Pyro-25 and Pyro-50, the 100 reflection peak was detected at $2\theta = 3.8°$, along with a small overlapping peak forming a tail at the high-angle side of the 100 peak. The samples synthesised with pyromellitic acid concentrations of 1–25 mol m$^{-3}$ contained DCPD. A certain amount of the DCPD used as the starting material remained in these samples after the reaction, as identified by PDF #00-011-0293. The reflection peaks identified as calcium pyromellitate were detected in the samples synthesised using pyromellitic acid concentrations of 25–100 mol m$^{-3}$. The XRD pattern of calcium pyromellitate

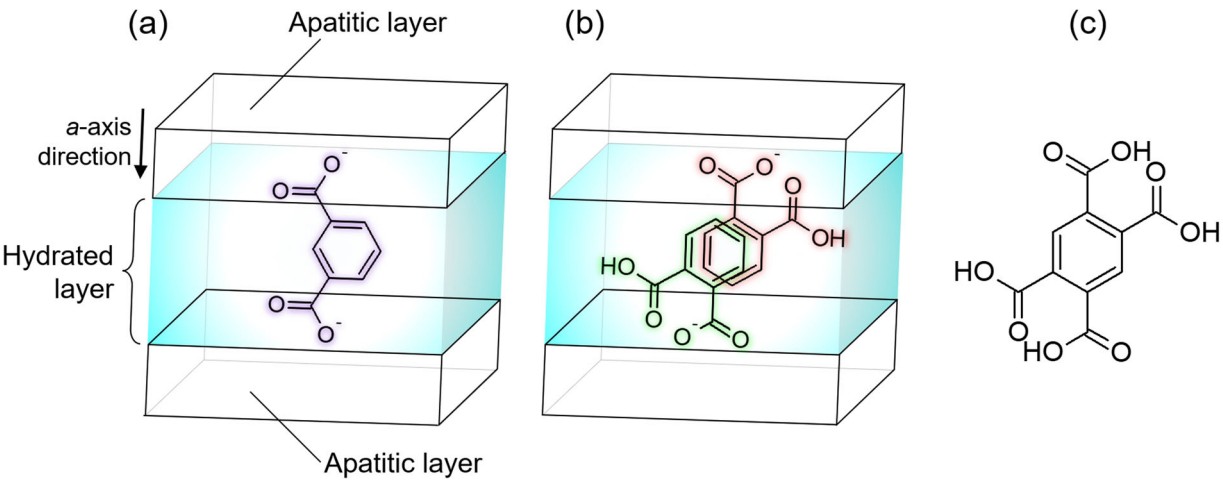

**Fig. 1 Schematic representation of interlayer structures and molecular structures of carboxylic acids.** Interlayer structures of OCP incorporated with **a** isophthalate ion, **b** phthalate ion dimer, and **c** molecular structure of pyromellitic acid.

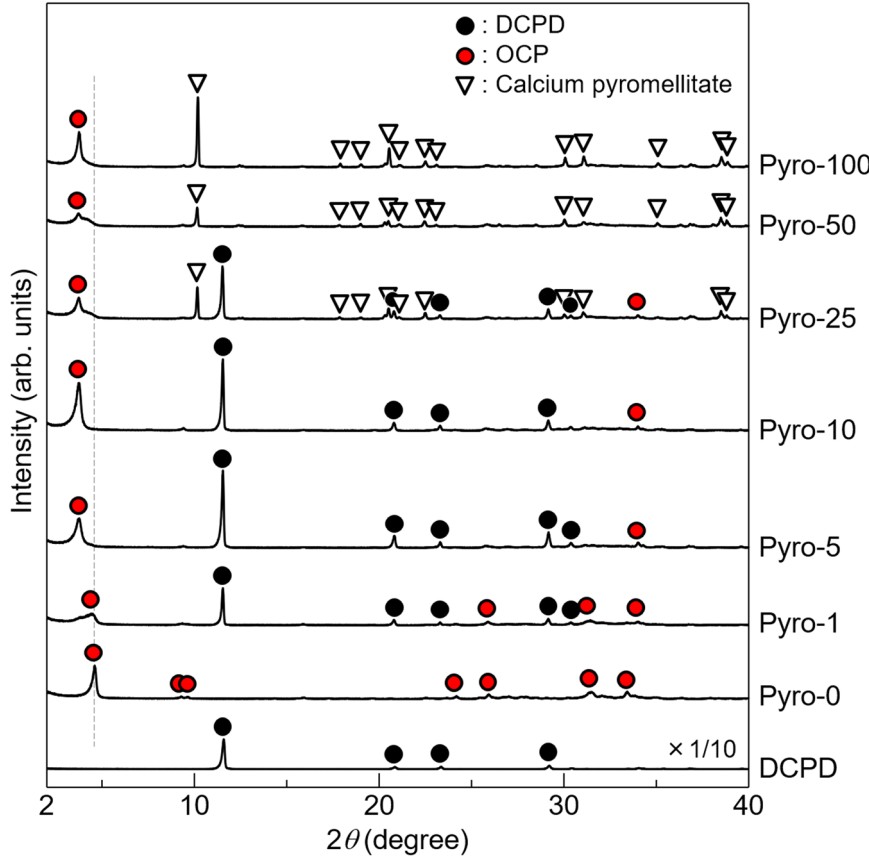

**Fig. 2 XRD analysis.** Powder XRD patterns of the samples synthesised hydrolytically from DCPD in a 200 mol m$^{-3}$ acetate buffer solution containing 0–100 mol m$^{-3}$ of pyromellitic acid with an initial pH of 5.5. The XRD pattern of DCPD is shown at one-tenth of its actual intensity.

was not available in the PDF database; therefore, calcium pyromellitate was identified based on the XRD pattern of calcium pyromellitate synthesised using a wet-chemical process (Supplementary Fig. 1).

Figure 2 demonstrates that a concentration of pyromellitic acid greater than 5 mol m$^{-3}$ is required to incorporate pyromellitate ions into OCP. However, single-phase OCP with incorporated pyromellitate ions could not be obtained by controlling the pyromellitic acid concentrations, as the dissociation states of pyromellitic acid depend on the reaction solution pH (as

explained in the Discussion section). Therefore, we investigated the effects of the reaction solution pH on the phase purity of OCP with incorporated pyromellitate ions.

**Optimising synthesis conditions: Effects of the reaction solution pH**. Powder XRD patterns of the samples synthesised in reaction solutions containing 5 mol m$^{-3}$ pyromellitic acid, with an initial pH varying from 5.0 to 7.0, are shown in Fig. 3. The samples were named in the form of Pyro-5_pH6.0, which

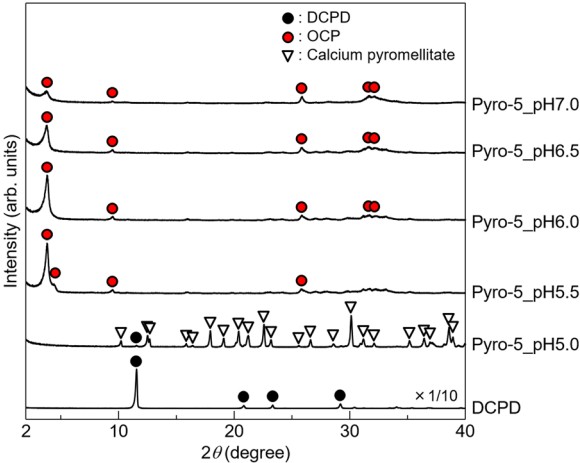

**Fig. 3 XRD analysis.** Powder XRD patterns of the samples synthesised hydrolytically from DCPD in a $200\,mol\,m^{-3}$ acetate buffer solution containing $5\,mol\,m^{-3}$ of pyromellitic acid with an initial pH ranging from 5.0 to 7.0. The XRD pattern of DCPD is shown at one-tenth of its actual intensity.

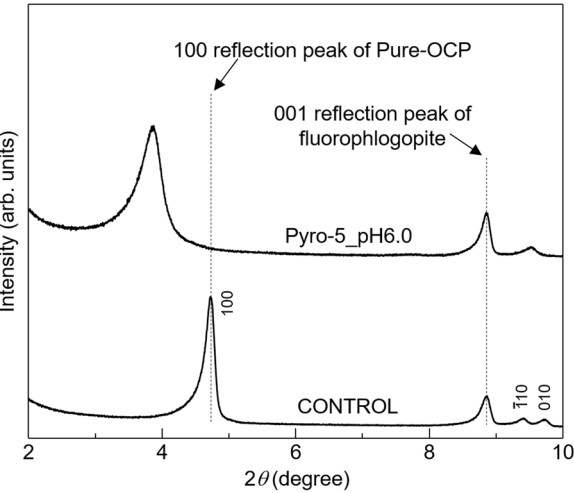

**Fig. 4 XRD analysis.** Powder XRD patterns of plain OCP (CONTROL) and OCP with incorporated pyromellitate ions (Pyro5_pH6.0). Fluorophlogopite powder was added to these samples as an angular standard material.

indicates the sample synthesised in the reaction solution containing $5\,mol\,m^{-3}$ pyromellitic acid with an initial pH of 6.0. In Pyro-5_pH5.0, calcium pyromellitate was formed, and a small amount of the DCPD precursor material remained in the sample. In Pyro-5_pH5.5, only the OCP phase was detected, while calcium pyromellitate was not. The 100 reflection peaks for OCP with incorporated pyromellitate ions and plain OCP were detected. DCPD was detected in Pyro-5 (Fig. 2), while it was not detected in Pyro-5_pH5.5 (Fig. 3). This difference was due to the DCPD amount being reduced for the pH study (as explained in the Methods section). Single-phase OCP with incorporated pyromellitate ions was formed in the Pyro-5_pH6.0, Pyro-5_pH6.5, and Pyro-5_pH7.0 samples. The intensities and sharpness of the 100 reflection peaks of these samples decreased with increasing pH, which was likely a result of the partial hydrolysis of OCP. Therefore, the conditions used to prepare the Pyro-5_pH6.0 sample were considered the most appropriate in this study, and further analyses of samples prepared under these conditions were performed.

**Chemical structures, compositions, and fluorescent properties.** We evaluated the crystal and chemical structures, composition, and fluorescent properties of the Pyro-5_pH6.0 sample. XRD analysis, with fluorophlogopite as an internal standard to calibrate the diffraction angle, was performed to precisely measure the $d_{100}$ value. Figure 4 shows the powder XRD patterns of OCP with incorporated pyromellitate ions (Pyro-5_pH6.0) and plain OCP, denoted as CONTROL. The CONTROL sample was synthesised using $CaCO_3$ and $H_3PO_4$ via a wet-chemical synthesis method[35]. According to the 100 reflection peak position, the $d_{100}$ value of OCP with incorporated pyromellitate ions (in Pyro-5_pH6.0) was calculated to be 2.29 nm, while that of the CONTROL was 1.87 nm. Therefore, due to the incorporation of the pyromellitate ions into the OCP interlayer, an increase in the interplanar spacing of 0.42 nm occurred.

The pyromellitate ions were incorporated by the substitution of hydrogen phosphate ions located in the OCP interlayer, as confirmed by chemical structural and compositional analyses. The chemical structures of the samples were characterised by Fourier-transform infrared (FTIR) spectroscopy. Figure 5 shows the FTIR spectra of the CONTROL, Pyro-5_pH6.0, and pyromellitic acid. We assigned the absorption peaks of the OCPs

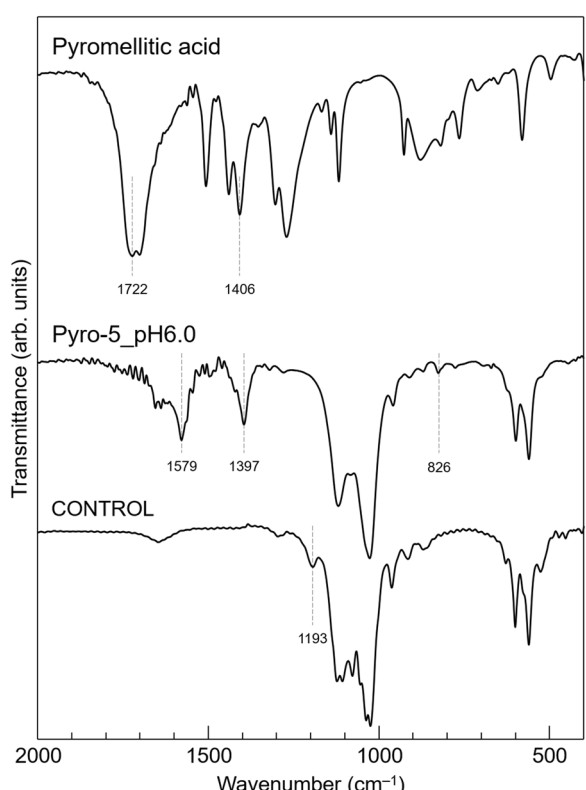

**Fig. 5 FTIR analysis.** FTIR spectra of plain OCP (CONTROL), OCP with incorporated pyromellitate ions (Pyro5_pH6.0), and pyromellitic acid obtained using the KBr pellet method.

based on previous reports[36,37]. In the CONTROL spectrum, the observed absorption peaks were assigned to plain OCP. Comparing the spectra of the CONTROL and Pyro-5_pH6.0, the absorption peaks at 1579, 1397, and $826\,cm^{-1}$ were only detected in Pyro-5_pH6.0. In the spectrum of Pyro-5_pH6.0, the absorption peaks at 1579 and $1397\,cm^{-1}$ were attributed to the dissociated carboxy groups ($-COO^-$) of the pyromellitate ions. On the other hand, the absorptions at 1722 and $1406\,cm^{-1}$, attributed to the carboxy groups ($-COOH$), were observed in the spectrum of pyromellitic acid, but were hardly detected in Pyro-

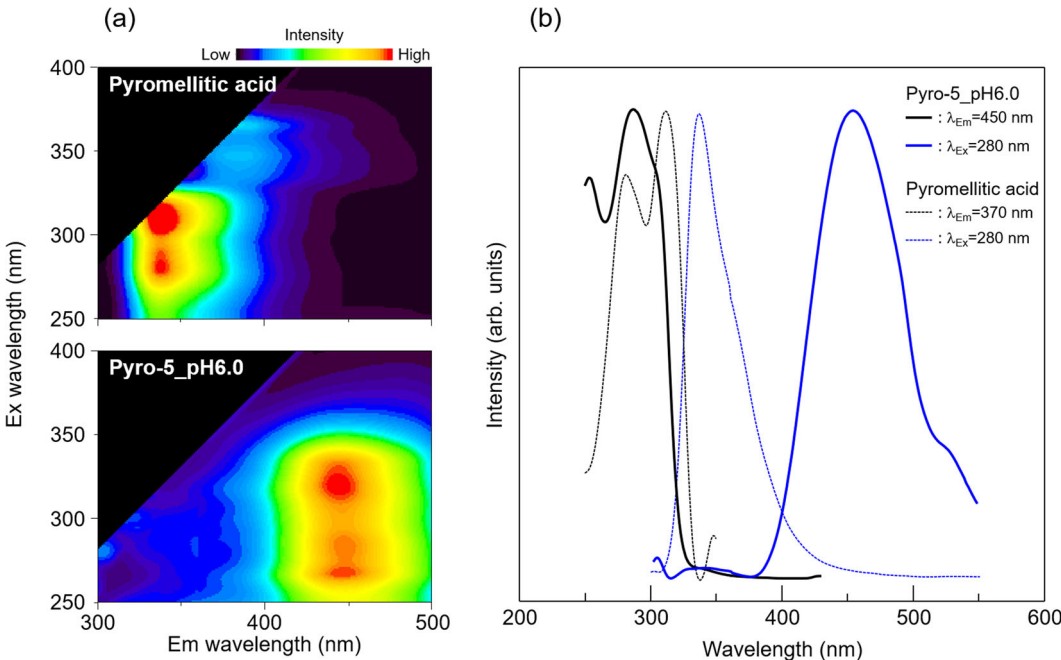

**Fig. 6 Fluorescence characterisation. a** 3D and **b** 2D fluorescent spectra of pyromellitic acid and OCP with incorporated pyromellitate ions (Pyro5_pH6.0). The intensities in these spectra were normalised.

5_pH6.0. Generally, the out-of-plane bending vibrations of C-H in 1,2,4,5-tetrasubstituted benzene are detected in the range of 900–850 $cm^{-1}$ [38]. Therefore, the small peak detected at 826 $cm^{-1}$ could be attributed to the out-of-plane bending vibrations of C–H in the benzene ring of the pyromellitate ions. The absorption peak at 1193 $cm^{-1}$, which was derived from the OH in-plane bending mode of $HPO_4^{2-}$ in the hydrated layer, was detected in the CONTROL, but not in Pyro-5_pH6.0.

The compositions, namely the Ca/P molar ratios, of the CONTROL and Pyro-5_pH6.0 were analysed using inductively coupled plasma atomic emission spectroscopy (ICP-AES) following the dissolution of the samples in an acidic solution. The Ca/P molar ratios of the CONTROL and Pyro-5_pH6.0 were 1.33 and 1.54, respectively. The Ca/P molar ratio of the CONTROL was equivalent to the stoichiometric composition of OCP (=1.33), while that of Pyro-5_pH6.0 was higher because of the substitution of the hydrogen phosphate ions by the pyromellitate ions.

3D fluorescent spectra, which are counter plots of the excitation (Ex) wavelength vs. emission (Em) wavelength vs. fluorescent intensity, of pyromellitic acid (powder) and OCP with incorporated pyromellitate ions (Pyro-5_pH6.0), are shown in Fig. 6a. In both 3D spectra, two peaks were observed. Pyromellitic acid exhibits strong fluorescence at 340 nm for Ex wavelengths of approximately 280 and 310 nm. After incorporation of the ions, strong fluorescence was detected at 450 nm for Ex wavelengths of 260 and 320 nm. OCP with incorporated pyromellitate ions exhibited a large Stokes shift of more than 100 nm. The 2D excitation and fluorescent spectra of pyromellitic acid and the OCP with incorporated pyromellitate ions (Pyro-5_pH6.0) are shown in Fig. 6b. The fluorescence spectra clearly indicate that the emission wavelength changed from 340 nm to 450 nm upon incorporation of the pyromellitate ions into the OCP structure. Figure 7 shows the images of plain OCP, pyromellitic acid, and OCP with incorporated pyromellitate ions under visible and UV light (wavelengths: 365, 312, and 254 nm). Purple emissions associated with pyromellitic acid were visible only at a UV wavelength of 312 nm and barely visible at other wavelengths. Brilliant blue emissions from OCP with incorporated pyromellitate ions (Pyro-5_pH6.0) were observed at UV wavelengths of 312 and 254 nm and were hardly visible at 365 nm. Plain OCP did not show characteristic emissions under UV light.

## Discussion

The experimental data suggested that pyromellitate ions were successfully incorporated into OCP. The XRD analyses (Figs. 2 and 3) confirmed that OCP was successfully produced with incorporated pyromellitate ions by precisely modifying the pyromellitic acid concentration and the pH of the reaction solution. When using high pyromellitic acid concentrations, single-phase OCP could not be obtained owing to the formation of poorly soluble calcium pyromellitate. Therefore, minimising the pyromellitic acid concentration was necessary to obtain single-phase OCP. Additionally, the pH of the reaction solution was an equally important parameter for obtaining single-phase OCP. As pyromellitic acid is a tetracarboxylic acid, it has complex dissociation states. The pH dependence on the dissociation states of pyromellitic acid was calculated based on the dissociation constants ($pK_{a1} = 1.92$, $pK_{a2} = 2.77$, $pK_{a3} = 4.36$, and $pK_{a4} = 5.35$)[39], which are shown in Supplementary Fig. 2. This figure shows that the trivalent and tetravalent pyromellitate ions ($HPy^{3-}$ and $Py^{4-}$) coexisted in the pH 5.5 reaction solution, and the molar fraction of $HPy^{3-}:Py^{4-}$ was approximately 2:3. The presence of pyromellitic acid as a single ionic species in the reaction solution is preferred for its incorporation into OCP, rather than coexisting as multiple ionic species. By examining the effects of the reaction solution pH, it was noted that the single-phase OCP was formed under higher pH conditions (>6.0) due to the dominant presence of a single ionic species, namely tetravalent pyromellitate ions, in the reaction solution.

In terms of phase purity and crystallinity, a pyromellitic acid concentration of 5 mol $m^{-3}$ at an initial pH of 6.0 was considered as the optimum synthesis condition for the experiments. The FTIR spectra (Fig. 5) also indicate the incorporation of pyromellitate ions into OCP. The absorption peaks derived from the

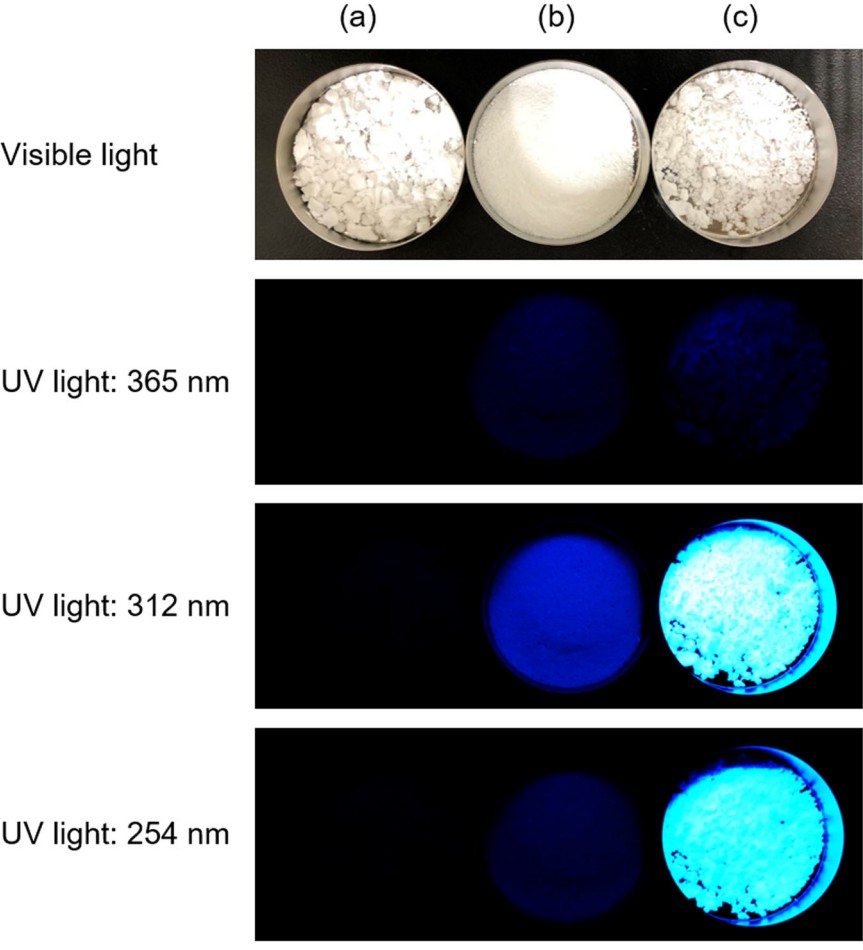

**Fig. 7 Images of samples under visible and UV light (365, 312, and 254 nm).** (a) Plain OCP (CONTROL), (b) pyromellitic acid, and (c) OCP with incorporated pyromellitate ions (Pyro5_pH6.0).

dissociated carboxy groups ($-COO^-$) and benzene rings of pyromellitic acid were observed in the spectrum of OCP with incorporated pyromellitate ions (Pyro-5_pH6.0). In addition, the absorption peak at $1193\,cm^{-1}$ derived from $HPO_4^{2-}$ in the hydrated layer was detected in the CONTROL sample, but not in the Pyro-5_pH6.0 sample. These findings suggested that although pyromellitic acid is a tetravalent carboxylic acid, the pyromellitate ions were successfully incorporated into the OCP interlayer by substitution of $HPO_4^{2-}$ in the hydrated layer by a similar manner as dicarboxylic acids. The fraction of $HPO_4^{2-}$ substituted by the pyromellitate ions was calculated using the Ca/P molar ratio of the sample. The compositional formula of OCP with incorporated $n$-valent carboxylate ions is expressed as $Ca_8(HPO_4)_{2-x}(n$-valent carboxylate ion$)_{2x/n}(PO_4)_4 \cdot mH_2O$ (in the case of pyromellitate ion; $2 \le n \le 4$), in which only one of the two hydrogen phosphate ions can be substituted[14]; therefore, the fraction of substitution, $x$, has a value between 0 and 1. As the Ca/P molar ratio of Pyro-5_pH6.0 was 1.54, $x$ was 0.81.

On incorporating the pyromellitate ions into the OCP interlayer, at least two carboxy groups of the pyromellitate ions should connect the apatitic layers (Fig. 1) because Ca-OOC(CH$_2$)$_n$COO-Ca (where the Ca ions are located in the apatitic layers) is the bonding structure proposed for the dicarboxylate ions located in the hydrated layers[14]. In addition, the $d_{100}$ value of OCP with incorporated dicarboxylate ions increased with increasing molecular chain length; therefore, the distance between the terminal carboxy groups of the incorporated dicarboxylate ions increased, especially in aliphatic dicarboxylic acids[32]. This occurred because

the main chain of the dicarboxylic acids was parallel to the $a$-axis of the OCP crystal lattice. Pyromellitic acid has four carboxy groups and there are three distances between the carboxy groups, but it is unclear which distance determines the $d_{100}$ value of OCP with incorporated pyromellitate ions. Therefore, we discuss here the contribution of the combination of the two carboxy groups for determining the $d_{100}$ value, namely those at the ortho, meta, and para positions of the pyromellitate ions. As a parameter that represents the molecular chain length of the incorporated dicarboxylate ions, we define $L$ as the distance between the carbon atoms of two carboxy groups in the carboxylic acid molecules, which was evaluated using quantum chemical calculation software. Supplementary Fig. 3 shows a schematic illustration of the parameter $L$. Figure 8a shows the relationship between the $d_{100}$ value of OCP with the incorporated aliphatic dicarboxylate ions, namely succinic, glutaric, adipic, and suberic ions, as well as the $L$ values of these carboxylic acids. The $d_{100}$ values of the OCP samples with incorporated succinic, glutaric, adipic, and suberic ions are 21.4, 22.3, 23.6, and 26.1 Å, respectively[32], corresponding to $L$ values of 3.86, 5.07, 6.40, and 8.95 Å, respectively. The figure indicates that the $d_{100}$ values of OCP with incorporated aliphatic dicarboxylate ions are proportional to the $L$ values of the dicarboxylic acids. Linear curve fitting of the data provided the following relationship:

$$d_{100} = 0.9355L + 17.669\ldots \qquad (1)$$

The correlation coefficient ($R^2$) of this linear expression was 0.9974, indicating a good fit to the data. As the crystal system of

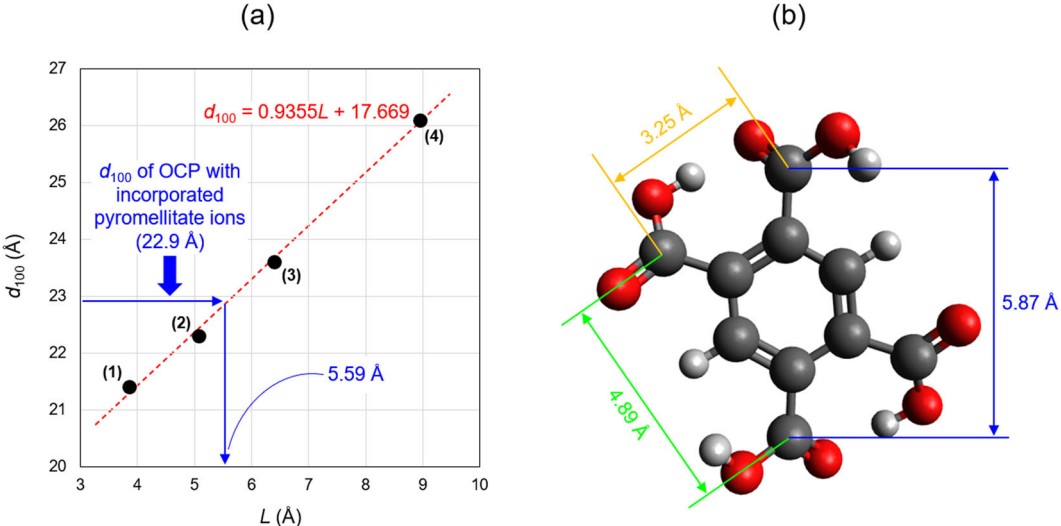

**Fig. 8 Incorporated molecular size and $d_{100}$ values of OCP. a** Relationship of the distance between the carbon atoms of two carboxy groups ($L$) of (1) succinic, (2) glutaric, (3) adipic, and (4) suberic acids, and the $d_{100}$ values of OCP incorporated with these acids. **b** Schematic illustration defining the various $L$ values of the pyromellitic acid.

OCP is triclinic, the $d_{100}$ value is not equal to the length of the $a$-axis. According to PDF #01-074-1301, the length of the $a$-axis and $d_{100}$ value of OCP are 19.87 Å and 18.78 Å, respectively. Therefore, the conversion factor from the length of the $a$-axis to the $d_{100}$ value was 0.9451 (=18.78/19.87). The coefficient of $L$ (0.9355) in Eq. (1) almost matched this conversion factor because the dicarboxylate ions were parallel to the $a$-axis in the OCP interlayer. The similarity between both values indicates that estimating $L$ from $d_{100}$ is viable. The $d_{100}$ value of OCP with incorporated pyromellitate ions was 22.9 Å (Fig. 4), giving an $L$ value of the pyromellitate ions in the OCP interlayer of 5.59 Å. Figure 8b shows that the $L$ values for the ortho, meta, and para carboxy groups of pyromellitic acid are 3.25, 4.89, and 5.87 Å, respectively. Therefore, based on the linear relationship found in the $d_{100}$ values of OCP with incorporated aliphatic dicarboxylate ions and their respective $L$ values, it was inferred that the distance of the carboxy groups at the para-position in pyromellitic acid mainly dominated the $d_{100}$ value of OCP. The above discussion does not take into account differences in the main chain structures (viz. methylene group or phenyl group) and valences of the incorporated carboxylate ions, but it may be necessary to consider these factors.

Incorporating pyromellitate ions into OCP imparted fluorescent properties to OCP. For pyromellitic acid, the intensity of the excitation spectrum in the wavelength range of 280 to 310 nm was higher than that at other wavelength ranges (Fig. 6a, b). Therefore, purple emissions from pyromellitic acid were observed strongly at a UV wavelength of 312 nm and were hardly visible at other UV wavelengths (Fig. 7). In contrast, the intensity of the excitation spectrum of OCP with incorporated pyromellitate ions (Pyro-5_pH6.0) was higher in the 260–320 nm range than in other wavelength ranges. Consequently, brilliant blue emissions were observed at wavelengths of 312 and 254 nm, with almost no emissions visible at 365 nm (Fig. 7). Because the wavelength for the maximum emission of pyromellitic acid (340 nm) slightly exceeds the lower limit of the wavelength range that can be visually observed, it is difficult to compare the actual emission intensities of pyromellitic acid and the OCP with incorporated pyromellitate ions based on Fig. 7. However, the emission intensity of OCP with incorporated pyromellitate ions was clearly higher than that of pyromellitic acid when observed using an optical filter known as an 'eye'. The development of a sensing

technique involving naked eye observations, such as advanced fluorescence diagnosis, using changes in the chemical states of incorporated carboxylate ions (in this case, pyromellitic acid), will be inspired by such clear differences in emission intensities.

Therefore, we successfully incorporated a tetravalent carboxylic acid, namely pyromellitic acid, into OCP, which had not been previously achieved. We estimated the structure of the interlayer pyromellitic acid by a computational approach and found that the meta- and para- positions of the carboxy groups of the pyromellitate ions likely bonds the apatitic layers. However, the details of its position in the OCP interlayer could not be clarified, and hence, our future research involving atomic-resolution structural analysis will address this point. The outcomes of this study are expected to significantly increase the types of carboxylic acids that can be incorporated into OCPs and expand their range of applications. In addition, we demonstrated that incorporating a carboxylic acid imparts fluorescent properties to OCP. Therefore, as OCP has a high affinity for hard tissue in the human body and can be used as a material for artificial bones, its fluorescent properties could be used in the development of a theranostic material enabling repair of bones and teeth and fluorescence diagnosis. OCP materials with incorporated carboxylate ions can provide unique functionalities that could enable the development of biofriendly functional materials for artificial bones, dental implants, biosensors, and theranostic applications in the future.

## Methods

**Chemicals**. Chemicals including calcium nitrate tetrahydrate ($Ca(NO_3)_2 \cdot 4H_2O$, 98%), calcium chloride ($CaCl_2$, 95.0%), diammonium hydrogen phosphate (($NH_4)_2HPO_4$, 98%), sodium hydroxide (NaOH, 97%), phosphoric acid ($H_3PO_4$, 85% aqueous solution), ammonia solution (25% aqueous solution), and acetic acid (99.7%), were purchased from FUJIFILM Wako Pure Chemical Corp., Osaka, Japan. Pyromellitic acid (98.0%) was purchased from Tokyo Chemical Industry Co., Ltd., Tokyo, Japan. Furthermore, calcium carbonate ($CaCO_3$ (calcite), 99.5%) and hydrochloric acid solution (HCl, 1.0 mol dm$^{-3}$) were purchased from Nacalai Tesque Inc., Kyoto, Japan. All chemicals were used without further purification.

**Synthesis of DCPD as a precursor of OCP**. We synthesised DCPD as a precursor of OCP using the following process developed by our research group[40]. A solution of calcium nitrate (100 cm$^3$, 200 mol m$^{-3}$) was stirred at 30 °C at 500 rpm in a glass beaker using a magnetic stirrer. Subsequently, a diammonium hydrogen phosphate solution (100 cm$^3$, 200 mol m$^{-3}$) was added to the calcium nitrate solution. A white precipitate was immediately formed after the two solutions were mixed. After

10 min, the formed precipitate was collected by vacuum filtration and washed with distilled water and ethanol, and then dried at 40 °C overnight.

**Synthesis of OCP with incorporated pyromellitate ions**. We synthesised OCP with incorporated pyromellitate ions by the hydrolysis of DCPD in an acetate buffer solution containing pyromellitic acid, based on the method developed by our research group with appropriate modifications[40]. To evaluate the effects of the pyromellitic acid concentrations, synthesised DCPD (1.38 g) was added to an acetate buffer solution (100 cm$^3$, 200 mol m$^{-3}$) containing 0–100 mol m$^{-3}$ pyromellitic acid at 60 °C under vigorous stirring. The initial pH of the reaction solution was adjusted to 5.5 using an ammonia solution. To evaluate the effects of the initial pH of the pyromellitic acid solution, the same process was used with a solution containing a fixed amount (5 mol m$^{-3}$) of pyromellitic acid. However, the amount of DCPD was decreased from 1.38 g to 0.344 g, as excess DCPD was observed in the XRD pattern of the prepared OCP after a reaction for 1 h in solutions containing 5 mol m$^{-3}$ pyromellitic acid (Fig. 2). The initial pH of the reaction solution was varied from 5.0 to 7.0. In both sets of experiments, after 1 h, the precipitate was collected by vacuum filtration and washed with distilled water and ethanol, and then dried at 40 °C overnight.

**Synthesis of the plain OCP control sample**. Plain OCP (CONTROL) was synthesised using calcium carbonate and phosphoric acid[35]. Calcium carbonate (8 mmol) and phosphoric acid (6 mmol) were mixed in ultrapure water (100 cm$^3$) at 60 °C via stirring. After 3 h, the pH of the calcium phosphate suspension was decreased to 5.0 by adding an appropriate amount of HCl. After an additional 30 min of stirring at 60 °C, the plain OCP precipitate was isolated by vacuum filtration and gently rinsed with ultrapure water and ethanol, followed by drying at 40 °C overnight.

**Synthesis of calcium pyromellitate for crystalline phase identification**. A calcium chloride solution (5 cm$^3$, 1000 mol m$^{-3}$) was added to an acetate buffer solution (200 mol m$^{-3}$) containing pyromellitic acid (5 mol m$^{-3}$) at 60 °C, under continuous stirring. The initial pH of the acetate buffer solution containing pyromellitic acid was adjusted to 5.5 using an ammonia solution. A white precipitate was formed immediately after adding the calcium chloride solution, which was assumed to be pure calcium pyromellitate. The obtained calcium pyromellitate was collected by vacuum filtration, washed with distilled water and ethanol, and then dried at 40 °C overnight.

**Characterisation**. The crystalline phases of the samples were characterised by powder XRD (RINT-2000, Rigaku, Tokyo, Japan) using Cu-Kα radiation. The $d_{100}$ values of the OCP samples were evaluated by powder XRD using Cu-Kα radiation in the range $2\theta = 2°$ to $10°$. For evaluation of the $d_{100}$ values, the sample powders were mixed with fluorophlogopite (Topy Industries Ltd., Tokyo, Japan) as an angular standard. The mass ratio of the sample to fluorophlogopite was 4:1. The chemical structures of the samples were characterised by FTIR (Frontier MIR/NIR, PerkinElmer Japan, Kanagawa, Japan) using the KBr pellet method. The fluorescent properties of the sample powder were characterised at room temperature (around 25 °C) using a fluorescence spectrometer (FP-8300, JASCO, Tokyo, Japan). The 3D fluorescent spectra were obtained by scanning with an Em wavelength ($\lambda_{Em}$) of 300–500 nm and Ex wavelength ($\lambda_{Ex}$) of 250–400 nm. The 2D excitation and fluorescent spectra of the pyromellitic acid powder were obtained under the conditions of $\lambda_{Ex} = 280$ nm and $\lambda_{Em} = 370$ nm, respectively. We selected an $\lambda_{Em}$ value slightly larger than the strongest fluorescent wavelength of pyromellitic acid (340 nm), to ensure a wider range of wavelengths for analysis. The 2D excitation and fluorescent spectra of the OCP with incorporated pyromellitate ions were obtained under the conditions of $\lambda_{Ex} = 280$ nm and $\lambda_{Em} = 450$ nm, respectively. The fluorescent behaviour of the samples was observed under UV light using UV lamps (VL-6.LC, Vilber Lourmat, Marne-la-Vallée, France and EB-160C/J, Spectronics Corp., N.Y., U.S.A.). The Ca/P molar ratios of the samples were determined using ICP-AES (ICP-8100, Shimadzu Co., Kyoto, Japan) following the dissolution of the sample powders in aqua regia.

**Computational details**. To discuss the geometry of the pyromellitate ions incorporated into the OCP, we used a computational approach. Calculations of all the optimised structures in the ground state (in a vacuum) were performed based on quantum chemical density functional theory using the Firefly program package (PC GAMESS) version 8.2.0[41]. The optimised structures for all the investigated carboxylic acids were calculated using the B3LYP method[42–45]. In addition, the 6-31 G (d) basis set was employed. The correspondence of the determined optimised geometric structures to the local minimum on the potential energy surface was confirmed by calculating the harmonic vibrational frequencies at the same calculation level, as well as by the absence of imaginary frequencies. The distance between the carbon atoms of the carboxy groups in the dicarboxylic acid with an optimised structure, namely the $L$ value, was measured using Avogadro Version 1.2.0 software (Supplementary Fig. 3)[46].

## Data availability

The data that support the findings of this study are available from the corresponding author upon reasonable request.

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

## Acknowledgements

We thank Ms. Akiko Nakashima of the Japan Fine Ceramics Center for technical support. This study was financially supported by JSPS KAKENHI (Grant Numbers: JP18K14308 and JP20H05181), and the Institute of Biomaterials and Bioengineering, Tokyo Medical and Dental University (Project: 'Creation of Life Innovation Materials for Inter-disciplinary and International Researcher Development' under the Ministry of Education, Culture, Sports, Science, and Technology, Japan). In addition, our work was supported by the Research Program for CORE lab of 'Dynamic Alliance for Open Innovation Bridging Human, Environment and Materials' under the 'Network Joint Research Center for Materials and Devices', and a research grant from The Murata Science Foundation.

## Author contributions

T.Y. conceived and designed the project, performed the experiments, and wrote the paper. T.G. and T. Sekino assisted with the experiments and data analysis. M.H. and T. Seki assisted with measurements of the fluorescent properties. M. Kamitakahara, C.O., S.K., S.T., and M. Kawashita advised on the synthesis of octacalcium phosphate with incorporated pyromellitate ions. All authors read and commented on the manuscript.

## Competing interests

The authors declare no competing interests.
