## [Peer Review File · Communications Chemistry]

Reviewers' comments:

Reviewer #1 (Remarks to the Author):

The study by Taishi Yokoi et al. has reported the synthesis and characterization of octacalcium phosphate (OCP) crystal incorporated with tetracarboxylic ions in the layered structure. The authors succeeded in the incorporation of pyromellitic ions in the hydrated layers on the basis of their previous finding of a type of dicarboxylic acids as a dimer in the OCP structure. The fluorescent property derived from the incorporation of pyromellitic ions endowed OCP-based materials with great potentials in the broader medical applications. This manuscript provides insights on the incorporation strategy of tetracarboxylic ions in the layered structure of OCP. Overall, this manuscript is well organized and meets the criteria of Communications Chemistry. This manuscript can be considered for publication after addressing the comments list as follows:

Specific comments:

-Page 4, paragraph 1: "This indicates that the incorporation of tetracarboxylate ions into OCP is prohibitively difficult". What are the possible reasons why the incorporation of the carboxylic acids with multiple valences is difficult? What are the main influence factors?

-Page 7, paragraph 1: "XRD pattern of calcium pyromellitate was not available in the PDF database; therefore, calcium pyromellitate was identified based on the XRD pattern of calcium pyromellitate synthesised using a wet-chemical process (Supplementary Fig. 1)." According to the pattern of calcium pyromellitate in Figure S1, the diffraction peaks of the patterns of the synthetic samples (Pyro-25, Pyro-50, and Pyro-100) did not match well.

-Why did the authors not use a uniform amount of DCPD in the synthesis process? According to the statement of the authors, they decreased the amount of DCPD from 1.38 g to 0.344 g. Why did the authors not set the amount of DCPD to 0.344 g in all the experiments?

-Why did the authors employ the OCP synthesized from calcium carbonate and phosphate acid as the control but not the sample produced by a similar method to the Pyro-incorporated OCP samples?

-Figure 5, FTIR spectra: why was the peak at 1579 cm^{-1} not detected in Pyro-5_pH6.0?

-Page 8, line 3 from the bottom: "829 cm^{-1} " should be corrected to "826 cm^{-1} " according to Figure 5.

-Page 11, paragraph 2: "These findings suggested that although pyromellitic acid is a tetravalent carboxylic acid, the sites occupied by the pyromellitate ions in the hydrated layer were the same as those of dicarboxylic acids." Please provide the relevant references or evidence.

Reviewer #2 (Remarks to the Author):

Review on:

COMMSCHEM-20-0364-T:

"Incorporation of tetracarboxylate ions into octacalcium phosphate for the development of next-generation biofriendly materials" by Taishi Yokoi et al.

General remarks:

The authors describe the synthesis of a new OCP phase with incorporated pyromellitate ions by the hydrolysis of dicalcium phosphate dihydrate $\text{CaHPO}_4 \cdot 2\text{H}_2\text{O}$ in the presence of pyromellitic acid. The product and reaction were registered by powder X-ray diffraction and IR spectroscopy. The pH was varied between 5 and 7.

Detailed remarks:

Page 6:

"Powder diffraction file (PDF) #01-074-1301."

Exists a corresponding citation or link to database?

Page 8:

"According to the 100 reflection peak position, the d100 value of the OCP with incorporated pyromellitate ions (in Pyro-5_pH6.0) was calculated to be 2.29 nm, while that of CONTROL was 1.87 nm. Therefore, an increase in the interplanar spacing of 0.42 nm occurred due to the incorporation of the pyromellitate ions into the OCP interlayer.

Increase from 18.7 to 22.9 would mean an increase of about 20% along a-direction. This is a quite large change and it would be interesting whether c and b dimensions are affected or not in order to estimate a volume increase. In Monma 1984 (citation 32) these values were not affected. How is it in this case?

Page 8:

Figure 5 shows the FTIR spectra of the CONTROL, Pyro-5_pH6.0, and pyromellitic acid. We assigned the absorption peaks of the OCPs based on a previous report^{36,37}.

Could the authors add as control a physical mixture (prepared from solution) of pyromellitic acid and OCP in KBr? The spectra of the single compounds is o.k., but the real control for me would be physical mixture against incorporated pyromellitic acid by chemical reaction. In this way one would see the effect of physisorbed pyromellitic acid on the OCP surface.

Page 10 and 11:

This figure shows that the trivalent and tetravalent pyromellitate ions (HPy³⁻ and Py⁴⁻) coexisted in the reaction solution with pH 5.5, and the molar fraction of HPy³⁻:Py⁴⁻ was approximately 2:3. The compositional formula of OCP with incorporated n-valent carboxylate ions is expressed as Ca₈(HPO₄)_{2-x}(n-valent carboxylate ion)_{2x/n}(PO₄)₄·mH₂O, in which only one of the two hydrogen phosphate ions can be substituted¹⁴; therefore, the fraction of substitution, x, has a value between 0 and 1. As the Ca/P molar ratio of Pyro-5_pH6.0 was 1.54, x was 0.81.

The authors state that in solution they have a trivalent and mostly tetravalent pyromellitate ions. The built in pyromellitate is divalent, substituting a divalent HPO₄ group. Can the authors explain the discrepancy of charges between the built-in (2-) and the free pyromellitate ions in the reaction solution (3-/4-)?

Page 12:

"Figure 8(a) shows the relationship between the d100 value of the OCP with the incorporated aliphatic dicarboxylate ions, namely succinic, glutaric, adipic, and suberic ions, as well as the L values of these carboxylic acids. The d100 values of the OCP samples with incorporated succinic, glutaric, adipic, and suberic ions are 21.4, 22.3, 23.6, and 26.1 Å, respectively³², corresponding to L values of 3.86, 5.07, 6.40, and 8.95 Å, respectively."

The authors use data of former measurements from 1984 by Monma (citation 32) in order to prove that there is a correlation between incorporated acid and OCP cell size. This is somehow problematic since the acids investigated in 1984 were divalent aliphatic acids with conformational freedom of the carbohydrate chain whereas the pyromellitate acid is aromatic thus rigid and has four acidic groups. Also, I am wondering how the OCP lattice can be extended from 18.7 Å up to over 26 Å (suberic acid) corresponding to 30% extension or 22.9 Å (pyromellitate acid) without destroying the structure by breaking the interconnecting Ca - HPO₄ bonds in the B-layer along a-direction. This would mean that HPO₄ in the A-layer is just replaced by the aromatic acid and the remaining motif is stretched without any impact on the crystal lattice.

Page 21 and 22:

In the Figs. 3 and 4 the X-rays reflections of OCP are indicated. It would be helpful to index them by their (hkl) value at least for Fig. 3 to see which peak belongs to which lattice plane.

These peaks could correspond to 010 and 002 which may indicate a change along b or c-axes.

Conclusion:

The work seems to be conclusive in a confined context and the authors seem to have success to synthesize a new "OCP-organic acid composite" variant. Nevertheless, it bears some inconsistencies on the general level.

The work is based on initiate work of Monma and others from 1984 and later. They set up the model with extended OCP-cell in a -direction and the built-in organic acid based on X-ray data. However, Manma et al. did not present any X-ray curve in the papers only the d-values of the registered lattice planes. The authors took over this interpretation and fit their data to these previous findings.

In my opinion, for a real proof a complete X-rays structure determination on the atomic scale would be necessary accompanied by NMR in order to show the composite nature of the new compound. However, looking on the X-rays where mainly only the 100 peak of OCP is really detectable, a quantitative evaluation at this state seems hopeless. This would need sample purification to get rid off byproducts and synthesis of larger amounts for synchrotron and neutrons. Thus, I would restrict my needs for corrections on the above indicated items.

Also, the authors could specify their figure 1. as Monma already did in 1984 by a sketch of the stretched atomic model of 001 zone of OCP with the inserted organic acid. Monma avoided to indicate the organic acid position inside OCP. There, they could indicate which HPO₄ group is replaced (B-layer). Then, it would become clear that along a-axis the HPO₄ group of A-layer is detached from vicinated calcium atoms by the stretching of 4 Å. On the other hand, a stretching is not needed because the distance between the HPO₄ groups of the B-layer in the apatitic region amounts to 7.5 Å from phosphorus to phosphorous in projection and 10 Å in total.

Thus, the simplistic model has its drawbacks and for future work only a professional and targeted structure resolution at atomic levels could solve the problem.

Also, the motivation of UV emission is not clear for me. Built-in bone replace material is not suitable for in vivo UV experiments. May be in teeth application it has more sense, where a surface damage is replaced.

Thus, I let the editors to decide whether the paper is suitable for publications in Comm. Chem., and if so then to accept after answering to the above questions.

Reviewer #3 (Remarks to the Author):

Major claims of the paper:

The major claim of the paper concerns an essential extension of the physical properties of octacalcium phosphate (OCP) by substitution of hydrogen phosphate ions in the hydrated layers of the crystal lattice with a fluorescent salt of a tetracarboxylic acid.

Novelty and interest to others in the community:

The presented results mean a significant progress based on first studies from H. Monma et al. in 1983/1984, and 2005. The authors succeeded to incorporate pyromellitic acid, a tetracarboxylic acid, in OCP for the first time. The functionalized OCP shows brilliant blue emission behavior under UV light. Such material is of high interest for fluorescence diagnostics of dental templates, and in a large field of orthopedic surgery.

Further influence on thinking in the field:

In the past OCP was only of interest as a precursor material for hard tissue repair. Now, these results could open the way to search for other materials in order to combine their basic medical functionality with sensor properties.

The researchers have reproduced their work in a straightforward way. All experimental details and related literature are given convincingly. Therefore, I propose the publication of the manuscript in this version.

Review on:**COMMSCHEM-20-0364-T:**

“Incorporation of tetracarboxylate ions into octacalcium phosphate for the development of next-generation biofriendly materials” by Taishi Yokoi et al.

General remarks:

The authors describe the synthesis of a new OCP phase with incorporated pyromellitate ions by the hydrolysis of dicalcium phosphate dihydrate $\text{CaHPO}_4 \cdot 2\text{H}_2\text{O}$ in the presence of pyromellitic acid. The product and reaction were registered by powder X-ray diffraction and IR spectroscopy. The pH was varied between 5 and 7.

Detailed remarks:**Page 6:**

“Powder diffraction file (PDF) #01-074-1301.”

Exists a corresponding citation or link to database?

Page 8:

“According to the 100 reflection peak position, the d_{100} value of the OCP with incorporated pyromellitate ions (in Pyro-5_pH6.0) was calculated to be 2.29 nm, while that of CONTROL was 1.87 nm. Therefore, an increase in the interplanar spacing of 0.42 nm occurred due to the incorporation of the pyromellitate ions into the OCP interlayer.

Increase from 18.7 to 22.9 would mean an increase of about 20% along a-direction. This is a quite large change and it would be interesting whether c and b dimensions are affected or not in order to estimate a volume increase. In Monma 1984 (citation 32) these values were not affected. How is it in this case?

Page 8:

Figure 5 shows the FTIR spectra of the CONTROL, Pyro-5_pH6.0, and pyromellitic acid. We assigned the absorption peaks of the OCPs based on a previous report^{36,37}.

Could the authors add as control a physical mixture (prepared from solution) of pyromellitic acid and OCP in KBr? The spectra of the single compounds is o.k., but the real control for me would be physical mixture against incorporated pyromellitic acid by chemical reaction. In this way one would see the effect of physisorbed pyromellitic acid on the OCP surface.

Page 10 and 11:

This figure shows that the trivalent and tetravalent pyromellitate ions (HPy³⁻ and Py⁴⁻) coexisted in the reaction solution with pH 5.5, and the molar fraction of HPy³⁻:Py⁴⁻ was approximately 2:3.

The compositional formula of OCP with incorporated n-valent carboxylate ions is expressed as Ca₈(HPO₄)_{2-x}(n-valent carboxylate ion)_{2x/n}(PO₄)₄·mH₂O, in which only one of the two hydrogen phosphate ions can be substituted¹⁴; therefore, the fraction of substitution, x, has a value between 0 and 1. As the Ca/P molar ratio of Pyro-5_pH6.0 was 1.54, x was 0.81.

The authors state that in solution they have a trivalent and mostly tetravalent pyromellitate ions. The built in pyromellitate is divalent, substituting a divalent HPO₄ group. Can the authors explain the discrepancy of charges between the built-in (2-) and the free pyromellitate ions in the reaction solution (3-/4-)?

Page 12:

“Figure 8(a) shows the relationship between the d₁₀₀ value of the OCP with the incorporated aliphatic dicarboxylate ions, namely succinic, glutaric, adipic, and suberic ions, as well as the L values of these carboxylic acids. The d₁₀₀ values of the OCP samples with incorporated succinic, glutaric, adipic, and suberic ions are 21.4, 22.3, 23.6, and 26.1 Å, respectively³², corresponding to L values of 3.86, 5.07, 6.40, and 8.95 Å, respectively.”

The authors use data of former measurements from 1984 by Monma (citation 32) in order to prove that there is a correlation between incorporated acid and OCP cell size. This is somehow problematic since the acids investigated in 1984 were divalent aliphatic acids with conformational freedom of the carbohydrate chain whereas the pyromellitate acid is aromatic thus rigid and has four acidic groups. Also, I am wondering how the OCP lattice can be extended from 18.7 Å up to over 26 Å (subertic acid) corresponding to 30% extension or 22.9 Å (pyromellitate acid) without destroying the structure by breaking the interconnecting Ca - HPO₄ bonds in the B-layer along a-direction. This would mean that HPO₄ in the A-layer is just replaced by the aromatic acid and the remaining motif is stretched without any impact on the crystal lattice.

Page 21 and 22:

In the Figs. 3 and 4 the X-rays reflections of OCP are indicated. It would be helpful to index them by their (hkl) value at least for Fig. 3 to see which peak belongs to which lattice plane.

These peaks could correspond to 010 and 002 which may indicate a change along b or c-axes.

Conclusion:

The work seems to be conclusive in a confined context and the authors seem to have success to synthesize a new "OCP-organic acid composite" variant. Nevertheless, it bears some inconsistencies on the general level.

The work is based on initiate work of Monma and others from 1984 and later. They set up the model with extended OCP-cell in a -direction and the built-in organic acid based on X-ray data. However, Manma et al. did not present any X-ray curve in the papers only the d-values of the registered lattice planes. The authors took over this interpretation and fit their data to these previous findings.

In my opinion, for a real proof a complete X-rays structure determination on the atomic scale would be necessary accompanied by NMR in order to show the composite nature of the new compound. However, looking on the X-rays where mainly only the 100 peak of OCP is really detectable, a quantitative evaluation at this state seems hopeless. This would need sample purification to get rid off byproducts and synthesis of larger amounts for synchrotron and neutrons. Thus, I would restrict my needs for corrections on the above indicated items.

Also, the authors could specify their figure 1. as Monma already did in 1984 by a sketch of the stretched atomic model of 001 zone of OCP with the inserted organic acid. Monma avoided to indicate the organic acid position inside OCP. There, they could indicate which HPO₄ group is replaced (B-layer). Then, it would become clear that along a-axis the HPO₄ group of A-layer is detached from vicinated calcium atoms by the stretching of 4 Å. On the other hand, a stretching is not needed because the distance between the HPO₄ groups of the B-layer in the apatitic region amounts to 7.5 Å **from phosphorus to phosphorous in projection and 10 Å in total.**

Thus, the simplistic model has its drawbacks and for future work only a professional and targeted structure resolution at atomic levels could solve the problem.

Also, the motivation of UV emission is not clear for me. Built-in bone replace material is not suitable for in vivo UV experiments. May be in teeth application it has more sense, where a surface damage is replaced.

Thus, I let the editors to decide whether the paper is suitable for publications in Comm. Chem., and if so then to accept after answering to the above questions.

Reply to reviewer's comments

Thank you for the valuable comments on our manuscript entitled “**Incorporation of tetracarboxylate ions into octacalcium phosphate for the development of next-generation biofriendly materials**,” which has been submitted to *Communications Chemistry*. We have revised the manuscript based on these comments. The revisions and our considerations are detailed below. The modified sections in the revised manuscript are highlighted in yellow.

We hope that the revised manuscript is acceptable for publication in *Communications Chemistry*.

Replies to the referees' comments and revisions to the manuscript

Reviewer #1:

The study by Taishi Yokoi et al. has reported the synthesis and characterization of octacalcium phosphate (OCP) crystal incorporated with tetracarboxylic ions in the layered structure. The authors succeeded in the incorporation of pyromellitic ions in the hydrated layers on the basis of their previous finding of a type of dicarboxylic acids as a dimer in the OCP structure. The fluorescent property derived from the incorporation of pyromellitic ions endowed OCP-based materials with great potentials in the broader medical applications. This manuscript provides insights on the incorporation strategy of tetracarboxylic ions in the layered structure of OCP. Overall, this manuscript is well organized and meets the criteria of Communications Chemistry. This manuscript can be considered for publication after addressing the comments list as follows:

1) Page 4, paragraph 1: “This indicates that the incorporation of tetracarboxylate ions into OCP is prohibitively difficult”. What are the possible reasons why the incorporation of the carboxylic acids with multiple valences is difficult? What are the main influence factors?

Reply:

Thank you for your comment. It is difficult to incorporate carboxylic acids with multiple valences because the calcium carboxylate salt forms, inhibiting the incorporation of the carboxylic acid.

The formation of the calcium carboxylate salt during the synthesis of OCP with an incorporated carboxylate makes it difficult to incorporate the carboxylate ion into the OCP structure, because the concentration of the carboxylate ion in the reaction solution is reduced by the formation of the calcium carboxylate salt. Since the carboxylate ion incorporation reaction competes with the incorporation of the hydrogen phosphate ions, the decrease in the carboxylate ion concentration is clearly

unfavourable to the incorporation of the carboxylate ion into the OCP. A typical carboxylic acid that is difficult to incorporate into OCP owing to the formation of a calcium carboxylate salt, is terephthalic acid [1]. Therefore, the development of a new synthetic route that can avoid the formation of calcium carboxylate salts is required for the synthesis of OCP with incorporated carboxylate ions.

In 2018, we developed such a new synthetic route for OCP with incorporated carboxylate ions [2]. The characteristic point of this synthetic route is the use of dicalcium phosphate dihydrate (DCPD, $\text{CaHPO}_4 \cdot 2\text{H}_2\text{O}$) as a starting material. α -Tricalcium phosphate (α -TCP, $\text{Ca}_3(\text{PO}_4)_2$) is generally used as a starting material in the conventional synthetic procedure for OCP with incorporated carboxylate ions [3]. Since the solubility of DCPD is lower than that of α -TCP at $\text{pH} < 10$ [4], i.e. the pH range for the OCP synthesis, DCPD is a preferred starting material for the synthesis of OCP with incorporated carboxylate ions, which form poorly soluble calcium salts. We tried to synthesise OCP with incorporated pyromellitate ions by using α -TCP as a starting material; however, this was unsuccessful due to the formation of the calcium pyromellitate salt. Meanwhile, we have successfully synthesised OCP with incorporated pyromellitate ions using DCPD as a starting material in this study. Therefore, we can conclude that the incorporation of carboxylic acids with multiple valences was difficult because of the formation of calcium salts with low solubility during the OCP synthesis.

Based on the above discussion, we modified the following sentence in the revised manuscript.

On page 4, lines 10–11 in the revised manuscript:

“This indicates that the incorporation of tetracarboxylate ions into OCP is prohibitively difficult; ...”

is revised as

“This indicates that the incorporation of tetracarboxylate ions into OCP is synthetically prohibitively difficult; ...”

[1] Yokoi, T., Kamitakahara, M., Kawashita, M. & Ohtsuki, C. Formation of organically modified octacalcium phosphate in solutions containing various amounts of benzenedicarboxylic acids. *J. Ceram. Soc. Jpn.* **121**, 219–225 (2013).

[2] Yokoi, T., Goto, T. & Kitaoka, S. Transformation of dicalcium phosphate dihydrate into octacalcium phosphate with incorporated dicarboxylate ions. *J. Ceram. Soc. Jpn.* **126**, 462–468 (2018).

[3] Example; Yamada, I. & Tagaya, M. Immobilization of 2,2'-bipyridine-5,5'-dicarboxylic acid in layered octacalcium phosphate. *Colloid Interface Sci. Commun.* **30**, 100182 (2019).

[4] Chow, L. C. Development of self-setting calcium phosphate cements. *J. Ceram.*

Soc. Jpn. **99**, 954–964 (1991).

2) Page 7, paragraph 1: “XRD pattern of calcium pyromellitate was not available in the PDF database; therefore, calcium pyromellitate was identified based on the XRD pattern of calcium pyromellitate synthesised using a wet-chemical process (Supplementary Fig. 1).” According to the pattern of calcium pyromellitate in Figure S1, the diffraction peaks of the patterns of the synthetic samples (Pyro-25, Pyro-50, and Pyro-100) did not match well.

Reply:

The peak intensities of the XRD pattern of calcium pyromellitate shown in Supplementary Fig. 1 are different from those in Figure 2, as the reviewer suggests; however, the peak positions are similar (please see the following figure showing a comparison of the XRD patterns of calcium pyromellitate and Pyro-100). Therefore, the peak assignment of calcium pyromellitate in Figure 2 is reasonable.

Figure. Comparison of XRD patterns of calcium pyromellitate and Pyro-100.

3) Why did the authors not use a uniform amount of DCPD in the synthesis process? According to the statement of the authors, they decreased the amount of DCPD from 1.38 g to 0.344 g. Why did the authors not set the amount of DCPD to 0.344 g in all the experiments?

Reply:

Thank you for your comment. We first studied the effects of the pyromellitic acid concentration on the formation of OCP with incorporated pyromellitate ions and next studied the effects of pH. In the first experiment using 1.38 g of DCPD, we found that OCP with incorporated pyromellitate ions was formed for Pyro-1, Pyro-5, and Pyro-10. However, the DCPD used as a starting material remained in these samples after one

hour of reaction (see Figure 2). Hence, we considered that 1.38 g of DCPD used for the synthesis was enough to form OCP with incorporated pyromellitate ions. Rather, from the perspective of the single-phase synthesis of OCP with incorporated pyromellitate ions, we thought that it would be desirable to reduce the amount of DCPD. Therefore, in the second experiment, we reduced the amount of DCPD from 1.38 g to 0.344 g and succeeded in obtaining a single phase of OCP with incorporated pyromellitate ions (see Figure 3).

The reasoning behind decreasing the amount of DCPD was simply explained in the manuscript as follows.

On page 16, lines 4–6 in the revised manuscript:

“However, the amount of DCPD was decreased from 1.38 g to 0.344 g, as excess DCPD was observed in the XRD pattern of the prepared OCP after a reaction for 1 h in solutions containing $5 \text{ mol} \cdot \text{m}^{-3}$ pyromellitic acid (Fig. 2).”

4) Why did the authors employ the OCP synthesized from calcium carbonate and phosphate acid as the control but not the sample produced by a similar method to the Pyro-incorporated OCP samples?

Reply:

The synthesis method for plain OCP from DCPD was proposed in 2016 [5] and is a relatively new method compared to that of OCP synthesized from calcium carbonate and phosphoric acid, which was proposed in 2006 [6]. Since not only our group but also other research groups [7] use the latter synthesis method, it was considered the more common synthetic method for OCP in the research field of calcium phosphate-related materials. Hence, we adopted it as the synthesis method for the CONTROL.

[5] Sugiura, Y., Onuma, K. & Yamazaki, A. Enhancement of $\text{HPO}_4\text{-OH}$ layered structure in octacalcium phosphate and its morphological evolution by acetic acid. *J. Ceram. Soc. Jpn.* 124, 1178–1184 (2016).

[6] Kamitakahara, M., Ohtsuki, C., Takahashi, A. & Tanihara, M. Effect of silane-coupling treatment on thermal decomposition of octacalcium phosphate. *J. Soc. Mater. Sci. Jpn.* **55**, 881–884 (2006).

[7] Example; Tuncer, M., Bakan, F., Gocmez, H. & Erdem, E. Capacitive behaviour of nanocrystalline octacalcium phosphate (OCP) ($\text{Ca}_8\text{H}_2(\text{PO}_4)_6 \cdot 5\text{H}_2\text{O}$) as an electrode material for supercapacitors: biosupercaps. *Nanoscale* **11**, 18375–18381 (2019).

5) Figure 5, FTIR spectra: why was the peak at 1579 cm⁻¹ not detected in Pyro-5_pH6.0?

Reply:

Since the peak at 1579 cm⁻¹ was detected in Pyro-5_pH6.0, we speculated that the reviewer asked us “why was the peak at 1579 cm⁻¹ not detected in pyromellitic acid?” or “why was the peak at 1722 cm⁻¹ not detected in Pyro-5_pH6.0?”. Assuming that our speculation is correct, please consider the following response.

The absorption peak at 1579 cm⁻¹ in Pyro-5_pH6.0 was attributed to dissociated carboxy groups (–COO⁻) of the pyromellitate ions. The absorption peak of the dissociated carboxy group (–COO⁻) was not detected in the FTIR spectrum of pyromellitic acid (C₆H₂(COOH)₄) because it does not contain dissociated carboxy groups (–COO⁻). It is reasonable that the absorption peak attributed to carboxy groups (–COOH) at 1722 cm⁻¹ was detected in pyromellitic acid, while the absorption peak of the carboxy groups (–COOH) was hardly detected in Pyro-5_pH6.0. These results imply that the pyromellitic acids had considerably dissociated states in the OCP interlayer.

The above explanation and discussion were described in the manuscript in the following section.

On page 8, line 14 to page 9, line 3 in the revised manuscript:

“The pyromellitate ions were incorporated by substitution of hydrogen phosphate ions located in the OCP interlayer, as confirmed by chemical structural and compositional analyses. The chemical structures of the samples were characterised by Fourier-transform infrared (FTIR) spectroscopy. Figure 5 shows the FTIR spectra of the CONTROL, Pyro-5_pH6.0, and pyromellitic acid. We assigned the absorption peaks of the OCPs based on previous reports^{36,37}. In the CONTROL spectrum, the observed absorption peaks were assigned to plain OCP. Comparing the spectra of the CONTROL and Pyro-5_pH6.0, the absorption peaks at 1579, 1397, and 826 cm⁻¹ were only detected in Pyro-5_pH6.0. In the spectrum of Pyro-5_pH6.0, the absorption peaks at 1579 and 1397 cm⁻¹ were attributed to the dissociated carboxy groups (–COO⁻) of the pyromellitate ions. On the other hand, the absorption peaks at 1722 and 1406 cm⁻¹, attributed to the carboxy groups (–COOH), were observed in the spectrum of pyromellitic acid, but were hardly detected in Pyro-5_pH6.0.”

On page 11, lines 7–9 in the revised manuscript:

“The absorption peaks derived from the dissociated carboxy groups (–COO⁻) and benzene rings of pyromellitic acid were observed in the spectrum of OCP with incorporated pyromellitate ions (Pyro-5_pH6.0).”

6) Page 8, line 3 from the bottom: “829 cm⁻¹” should be corrected to “826 cm⁻¹” according

to Figure 5.

Reply:

Thank you for your kind comment. We have corrected the wavenumber in the revised manuscript on page 8 line 21.

On page 8, lines 20–21 in the revised manuscript:

“..., the absorption peaks at 1579, 1397, and 829 cm^{-1} were only detected in Pyro-5_pH6.0.”
is revised as

“..., the absorption peaks at 1579, 1397, and 826 cm^{-1} were only detected in Pyro-5_pH6.0.”

7) Page 11, paragraph 2: “These findings suggested that although pyromellitic acid is a tetravalent carboxylic acid, the sites occupied by the pyromellitate ions in the hydrated layer were the same as those of dicarboxylic acids.” Please provide the relevant references or evidence.

Reply:

Thank you for your important comment. The dicarboxylate ions are likely to occupy the prescribed sites in the OCP interlayer [8,9]. The FTIR spectra shown in Figure 5 indicate that HPO_4^{2-} in the OCP interlayer was replaced by pyromellitate ions in a similar manner to dicarboxylic acids. However, their positions in the OCP interlayer cannot be decided based only on the FTIR results as this is still a controversial concept. In light of the reviewer’s comment we revised our manuscript as follows.

On page 11, lines 11–14 in the revised manuscript:

“These findings suggested that although pyromellitic acid is a tetravalent carboxylic acid, the sites occupied by the pyromellitate ions in the hydrated layer were the same as those of dicarboxylic acids.”

is revised as

“These findings suggested that although pyromellitic acid is a tetravalent carboxylic acid, the pyromellitate ions were successfully incorporated into the OCP interlayer by substitution of HPO_4^{2-} in the hydrated layer by a similar manner as dicarboxylic acids.”

[8] Monma, H. & Goto, M. Complexes of apatitic layered compound $\text{Ca}_8(\text{HPO}_4)_2(\text{PO}_4)_4 \cdot 5\text{H}_2\text{O}$ with dicarboxylates. *J. Inclu. Phenom.* **2**, 127–134 (1984).

[9] Mathew, M. & Brown, W. E. A structural model for octacalcium phosphate-succinate double salt. *Bull. Chem. Soc. Jpn.* **60**, 1141–1143 (1987).

Reviewer #2:

The authors describe the synthesis of a new OCP phase with incorporated pyromellitate ions by the hydrolysis of dicalcium phosphate dihydrate $\text{CaHPO}_4 \cdot 2\text{H}_2\text{O}$ in the presence of pyromellitic acid. The product and reaction were registered by powder X-ray diffraction and IR spectroscopy. The pH was varied between 5 and 7.

1) Page 6:

“Powder diffraction file (PDF) #01-074-1301.”

Exists a corresponding citation or link to database?

Reply:

Thank you for your comment. Based on some papers published in the Nature Publishing Group [10], it seems sufficient to indicate the powder diffraction file card number for readers and a citation is unnecessary.

[10] Example: Lu, B. -Q., Willhammar, T., Sun B. -B., Hedin, N., Gale J. D. & Gebauer, D. Introducing the crystalline phase of dicalcium phosphate monohydrate. *Nat. Commun.* **11**, 1546 (2020).

2) Page 8:

“According to the 100 reflection peak position, the d_{100} value of the OCP with incorporated pyromellitate ions (in Pyro-5_pH6.0) was calculated to be 2.29 nm, while that of CONTROL was 1.87 nm. Therefore, an increase in the interplanar spacing of 0.42 nm occurred due to the incorporation of the pyromellitate ions into the OCP interlayer.”

Increase from 18.7 to 22.9 Å would mean an increase of about 20% along a-direction. This is a quite large change and it would be interesting whether c and b dimensions are affected or not in order to estimate a volume increase. In Monma 1984 (citation 32) these values were not affected. How is it in this case?

Reply:

Thank you for your comment. We could not evaluate the length of the *b*- and *c*-axis of OCP with incorporated pyromellitate ions because of the following reason: In the case of OCP, the unit cell becomes larger, specifically in the *a*-axis direction, by carboxylate ion incorporation, and thus the shift of the XRD peaks is much more complicated than when the unit cell size changes isotropically. In addition, no powder diffraction file of OCP with incorporated carboxylate ions has been reported, making it difficult to index reflection peaks except for the 100 reflection. (Note that we indexed the reflection peak detected at $2\theta = 3.9^\circ$ as the 100 reflection, because the reflection peak of OCP with incorporated carboxylate ions detected at such a reflection angle

has been indexed as the 100 reflection in several papers [11].)

Therefore, in this paper, we could not mention the changes in the *b*- and *c*-axis by pyromellitate ion incorporation into OCP. In my personal opinion, as with the incorporation of other carboxylate ions [12], there is probably little change in the *b*- and *c*-axis by pyromellitate ion incorporation.

[11] Example: Davies, E. et al. Citrate bridges between mineral platelets in bone. *Proc. Natl. Acad. Sci. U. S. A.*, **111**, E1354–E1363 (2014).

[12] Monma, H. The incorporation of dicarboxylates into octacalcium bis(hydrogenphosphate) tetrakis(phosphate) pentahydrate. *Bull. Chem. Soc. Jpn.*, **57**, 599–600 (1984).

3) Page 8:

“Figure 5 shows the FTIR spectra of the CONTROL, Pyro-5_pH6.0, and pyromellitic acid. We assigned the absorption peaks of the OCPs based on a previous report(36, 37).”

Could the authors add as control a physical mixture (prepared from solution) of pyromellitic acid and OCP in KBr? The spectra of the single compounds is o.k., but the real control for me would be physical mixture against incorporated pyromellitic acid by chemical reaction. In this way one would see the effect of physisorbed pyromellitic acid on the OCP surface.

Reply:

Thank you for your comment. We tried to prepare the physical mixture of plain OCP and pyromellitic acid from solution; however, our trial was unsuccessful because of the OCP to DCPD crystalline phase change owing to the acidic property of the pyromellitic acid solution. Therefore, we evaluated the FTIR spectrum of a physical mixture of plain OCP and pyromellitic acid (plain OCP:pyromellitic acid = 1:1 (mass ratio)). The FTIR spectra including the physical mixture sample are shown in the following figure.

Figure. FTIR spectra of Pyro-5_pH6.0, a mixture of plain OCP and pyromellitic acid, plain OCP (CONTROL), and pyromellitic acid.

Almost all the absorption peaks in the FTIR spectrum of the mixture of plain OCP and pyromellitic acid corresponded to those of plain OCP or pyromellitic acid (please see the red dotted line in the figure); however, the absorption peaks at 1613 and 1252 cm^{-1} were not observed in those of plain OCP or pyromellitic acid. The absorption peaks at 1613 and 1252 cm^{-1} were likely derived from the C=O stretching and C-O stretching respectively, and they might appear because of interactions between the carboxy groups of pyromellitic acid and Ca^{2+} exposed on the OCP crystal surface. Therefore, there is a possibility that these absorption peaks are derived from adsorbed pyromellitic acid on the OCP crystal surface. However, these peaks were not observed in Pyro-5_pH6.0. Hence, we concluded that the effects of adsorbed pyromellitic acid on the OCP crystal surface of Pyro-5_pH6.0 in the FTIR spectrum was negligible and there is no need to include the FTIR spectrum of the mixture of plain OCP and pyromellitic acid in the manuscript.

4) Page 10 and 11:

“This figure shows that the trivalent and tetravalent pyromellitate ions (HPy^{3-} and Py^{4-}) coexisted in the reaction solution with pH 5.5, and the molar fraction of $\text{HPy}^{3-}:\text{Py}^{4-}$ was approximately 2:3.”

“The compositional formula of OCP with incorporated n -valent carboxylate ions is expressed as $\text{Ca}_8(\text{HPO}_4)_{2-x}(\text{n-valent carboxylate ion})_{2x/n}(\text{PO}_4)_4\text{mH}_2\text{O}$, in which only one of the two hydrogen phosphate ions can be substituted¹⁴; therefore, the fraction of substitution,

x, has a value between 0 and 1. As the Ca/P molar ratio of Pyro-5_pH6.0 was 1.54, x was 0.81.”

The authors state that in solution they have a trivalent and mostly tetravalent pyromellitate ions. The built in pyromellitate is divalent, substituting a divalent HPO₄ group. Can the authors explain the discrepancy of charges between the built-in (2-) and the free pyromellitate ions in the reaction solution (3-/4-)?

Reply:

Thank you for your comment. We apologize for misleading the reviewer. Although the reviewer believes that the pyromellitate ions in the OCP interlayer were divalent, we did not have conclusive evidence as to which, divalent, trivalent, or tetravalent, pyromellitate ion was predominantly present. Based on a previous study [13], we assumed that the pyromellitate ions in the OCP interlayer were divalent or higher. Hence the chemical formula was described as *n*-valent carboxylate ions, avoiding specification of the valence state of the pyromellitate ion. The reason behind the possible confusion is likely that the range of *n* values was not specified in the manuscript. Hence, we modified the following section in the revised manuscript.

On page 11, lines 14–16 in the revised manuscript:

“The compositional formula of OCP with incorporated *n*-valent carboxylate ions is expressed as Ca₈(HPO₄)_{2-*x*}(*n*-valent carboxylate ion)_{2*x*/*n*}(PO₄)₄·*m*H₂O, ...”

is revised as

“The compositional formula of OCP with incorporated *n*-valent carboxylate ions is expressed as Ca₈(HPO₄)_{2-*x*}(*n*-valent carboxylate ion)_{2*x*/*n*}(PO₄)₄·*m*H₂O (in the case of pyromellitate ion; 2 ≤ *n* ≤ 4), ...”

[13] Example: Monma, H. & Goto, M. Complexes of apatitic layered compound Ca₈(HPO₄)₂(PO₄)₄·5H₂O with dicarboxylates. *J. Inclu. Phenom.* **2**, 127–134 (1984).

5) “Figure 8(a) shows the relationship between the d₁₀₀ value of the OCP with the incorporated aliphatic dicarboxylate ions, namely succinic, glutaric, adipic, and suberic ions, as well as the L values of these carboxylic acids. The d₁₀₀ values of the OCP samples with incorporated succinic, glutaric, adipic, and suberic ions are 21.4, 22.3, 23.6, and 26.1 Å, respectively³², corresponding to L values of 3.86, 5.07, 6.40, and 8.95 Å, respectively.”

The authors use data of former measurements from 1984 by Monma (citation 32) in order to prove that there is a correlation between incorporated acid and OCP cell size. This is somehow problematic since the acids investigated in 1984 were divalent aliphatic acids with conformational freedom of the carbohydrate chain whereas the pyromellitate acid is aromatic thus rigid and has four acidic groups. Also, I am wondering how the OCP lattice

can be extended from 18.7 Å up to over 26 Å (suberic acid) corresponding to 30% extension or 22.9 Å (pyromellitate acid) without destroying the structure by breaking the interconnecting Ca-HPO₄ bonds in the B-layer along a-direction. This would mean that HPO₄ in the A-layer is just replaced by the aromatic acid and the remaining motif is stretched without any impact on the crystal lattice.

Reply:

Thank you for your comment. We consider it reasonable that the interlayer distance of a layered compound has a proportional relationship with the size of the incorporated molecules. We found a linear relationship between the d_{100} values of OCP with incorporated aliphatic dicarboxylate ions and their respective L values. However, as the reviewer suggested, there is a possibility that differences in the main chain structures (namely methylene group or phenyl group) and valences of the incorporated carboxylate ions affect the relationship, and thus we modified the following section in the revised manuscript as follows.

On page 13, lines 9–14 in the revised manuscript:

“Therefore, it was inferred that the distance of the carboxy groups at the para position in pyromellitic acid mainly dominated the d_{100} value of OCP.”

is revised as

“Therefore, based on the linear relationship found in the d_{100} values of OCP with incorporated aliphatic dicarboxylate ions and their respective L values, it was inferred that the distance of the carboxy groups at the para-position in pyromellitic acid mainly dominated the d_{100} value of OCP. The above discussion does not take into account differences in main chain structures (viz. methylene group or phenyl group) and valences of incorporated carboxylate ions, but it may be necessary to consider these factors.”

In addition, the reviewer had a question about the large interplanar spacing expansion of OCP with incorporated suberate ions. However, not only Monma (Ref. 32), but also other researchers, reported that such large interplanar spacing expansions occurred by the incorporation of suberate ions [14,15,16]. Therefore, we believe that the d_{100} values described in Ref. 32 are reliable, although the detailed mechanism of such large interplanar spacing expansions without destroying the crystal structure of OCP is unclear.

[14] Marković, M., Fowler, B. O. & Brown, W. E. Octacalcium phosphate carboxylates. 2. Characterization and structural considerations. *Chem. Mater.* **5**, 1406–1416 (1993).

[15] Kamitakahara, M., Okano, H., Tanihara, M. & Ohtsuki, C. Synthesis of octacalcium phosphate intercalated with dicarboxylate ions from calcium carbonate and phosphoric acid. *J. Ceram. Soc. Jpn.* **116**, 481–485 (2008).

[16] Yokoi, T., Kato, H., Kim, I. Y., Kikuta, K., Kawashita, M. & Ohtsuki, C. Synthesis of octacalcium phosphate with incorporated succinate and suberate ions. *Ceram. Int.* **38**, 3815–3820 (2012).

6) Page 21 and 22:

In the Figs. 3 and 4 the X-rays reflections of OCP are indicated. It would be helpful to index them by their (hkl) value at least for Fig. 3 to see which peak belongs to which lattice plane.

These peaks could correspond to 010 and 002 which may indicate a change along b or c-axes.

Reply:

Thank you for your comment. We used the powder diffraction file (PDF) #01-074-1301 to identify “plain” OCP. As we already explained in response to question 2 of reviewer #2, the shift behaviour of the XRD peaks is much more complicated than when the unit cell size changes isotropically. To avoid confusing the reader, we thought it better not to include the hkl index in Figure 3. Whereas, in the XRD pattern of the CONTROL in Figure 4 there should be no confusion by the readers even if the hkl indices (100, -110, and 010 reflections detected at 4.7, 9.3, and 9.7°, respectively) are described. Thus, they are described based on the PDF#01-074-1301.

7) Conclusion:

The work seems to be conclusive in a confined context and the authors seem to have success to synthesize a new “OCP-organic acid composite” variant. Nevertheless, it bears some inconsistencies on the general level.

The work is based on initiate work of Monma and others from 1984 and later. They set up the model with extended OCP-cell in a -direction and the built-in organic acid based on X-ray data. However, Manma et al. did not present any X-ray curve in the papers only the d-values of the registered lattice planes. The authors took over this interpretation and fit their data to these previous findings.

In my opinion, for a real proof a complete X-rays structure determination on the atomic scale would be necessary accompanied by NMR in order to show the composite nature of the new compound. However, looking on the X-rays where mainly only the 100 peak of OCP is really detectable, a quantitative evaluation at this state seems hopeless. This would need sample purification to get rid off byproducts and synthesis of larger amounts for synchrotron and neutrons. Thus, I would restrict my needs for corrections on the above indicated items.

Also, the authors could specify their figure 1. as Monma already did in 1984 by a sketch of the stretched atomic model of 001 zone of OCP with the inserted organic acid. Monma avoided to indicate the organic acid position inside OCP. There, they could indicate which HPO_4 group is replaced (B-layer). Then, it would become clear that along a-axis the HPO_4 group of A-layer is detached from vicinated calcium atoms by the stretching of 4 Å. On the other hand, a stretching is not needed because the distance between the HPO_4 groups of the B-layer in the apatitic region amounts to 7.5 Å from phosphorus to phosphorous in projection and 10 Å in total.

Thus, the simplistic model has its drawbacks and for future work only a professional and targeted structure resolution at atomic levels could solve the problem.

Reply:

We would like to express our sincere gratitude to the reviewers for understanding the position of our research in the research history of OCP with incorporated carboxylate ions and for providing appropriate advice for our future research.

As the reviewer probably knows, the atomic coordinates of OCP crystals containing carboxylate ions have not been reported. However, it is possible to estimate the approximate steric structures of the carboxylate ions in the OCP interlayer from the relationship between its molecular sizes and the interplanar spacing of the (100) plane of OCP with incorporated carboxylate ions by a computational approach, as discussed in this paper using Figure 8. Unfortunately, a more detailed discussion of the bonding between the carboxylate ions and apatitic layer of OCP is difficult through this approach, because of insufficient information regarding the atomic coordinates of the OCP crystals containing the carboxylate ions. We would like to emphasise that this approach to estimate the steric structures of the carboxylate ions in the OCP interlayer is our original results and reported for the first time in this paper. As the reviewers indicate, this approach is still incomplete with regards to understanding the bonding, but we consider that this simple approach has novelty and is useful for estimating the steric structures.

We agree with the reviewer's comment that structure resolution at atomic levels is necessary. Since it is difficult to immediately conduct structural analysis of our samples using synchrotron radiation facilities, we consulted with a technician in Rigaku Corp. (a top share manufacturer of X-ray diffraction equipment in Japan) about the possibility to analyse the crystal structure of our samples at an atomic level using laboratory-level X-ray equipment of Rigaku Corp. The Rigaku technician has a single crystal structure analysis technique which uses a micrometre-level fine crystal, and this was considered as a possibility to analyse the crystal structure of our sample. However, based on the scanning electron microscopy (SEM) images of our samples (see below figure), it was unfortunately established that the crystals were too thin to analyse their crystal structure. At this stage the crystal structure analysis of the

crystals is difficult; however, we understand the importance of crystal structure analysis and hence we would like to perform crystal structure analysis using not only experimental methods, including NMR analysis and synchrotron radiation structure analysis, but also through advanced computational approaches in our future research.

Figure. SEM images of our samples; (a) CONTROL (Plain OCP) and (b) Pyro-5_pH6.0 (OCP with incorporated pyromellitate ions).

Based on the above discussion, we revised our manuscript as follows.

On page 14, lines 10–14 in the revised manuscript:

“We estimated the structure of the interlayer pyromellitic acid by a computational approach and found that the meta- and para-positions of the carboxy groups of the pyromellitate ions likely bonds the apatitic layers. However, details of its position in the OCP interlayer could not be clarified, and hence, this point is an issue for future research.”

was added.

8) (Continue) Conclusion:

Also, the motivation of UV emission is not clear for me. Built-in bone replace material is not suitable for in vivo UV experiments. May be in teeth application it has more sense, where a surface damage is replaced.

Reply:

Thank you for your comment. As the reviewer indicates, we know that UV light is too energetic as an excitation source for *in vivo* imaging. Our research article is not

just to present our research results, but also to inspire biomaterial researchers. We believe that OCP is a promising next generation biomaterial because of its incorporation property, and we have shown the results of the fluorescence experiments in the hope that the development of functional biomaterials using OCP will progress. We believe that the reviewer agrees with our idea.

We did not mention teeth repairing applications in our original manuscript, but, as the reviewer indicates, it is a potential application, therefore we modified the following section in the revised manuscript.

On page 14, lines 18–19 in the revised manuscript:

“fluorescent properties could be used in the development of a theranostic material enabling bone repair and ...”

is revised as

“fluorescent properties could be used in the development of a theranostic material enabling repair of bones and teeth and fluorescence diagnosis ...”

REVIEWERS' COMMENTS:

Reviewer #1 (Remarks to the Author):

The revised manuscript can be considered for acceptance.

Reviewer #2 (Remarks to the Author):

25th Nov 2020

Review on:

COMMSCHEM-20-0364-T:

"Incorporation of tetracarboxylate ions into octacalcium phosphate for the development of next-generation biofriendly materials" by Taishi Yokoi et al.

Second run of revision:

The authors answered point by point my questions and modified the manuscript accordingly.

Also, I like to thank for the interesting papers/citations which I did not know concerning a newly detected CaP phase and the presence of a citrate-OCP composite in bone.

Additionally, I acknowledge their effort to try an X-ray structure determination in cooperation with Rikagu company. May be, in future it will be possible to resolve the structure on the thin plates by 3D electron diffraction techniques such as given by the authors: Lu, et al. Nat. Commun. 11, 1546 (2020). There are several groups who are able to do this such as the Palatinus group in Prague or Sven Hovmöller in Stockholm. Ute Kolb in Mainz is specialized on organics structure determination by electron diffraction as well Mauro Gemmi in Pisa, Italy.

Thus, I recommend the paper to be published.